# For the Parent, by the Parent: Creating a Program to Empower Parents of Refugee Background in Canada Using Novel Participatory Approaches

**DOI:** 10.3390/children9121816

**Published:** 2022-11-24

**Authors:** Pardeep Kaur Benipal, Bernice Ho, Tanvir Kaukab, Meb Rashid, Ashna Bowry, Aisha K. Yousafzai, Ripudaman Singh Minhas

**Affiliations:** 1Department of Pediatrics, St. Michael’s Hospital, Unity Health Toronto, Toronto, ON M5C 2T2, Canada; 2Temerty Faculty of Medicine, University of Toronto, Toronto, ON M5S 1A8, Canada; 3Women’s College Hospital, Toronto, ON M5S 1B2, Canada; 4Li Ka Shing Knowledge Institute, St. Michael’s Hospital, Unity Health Toronto, Toronto, ON M5B 1W8, Canada; 5T.H. Chan School of Public Health, Harvard University, Boston, MA 02138, USA; 6Division of Developmental Pediatrics, Department of Paediatrics, University of Toronto, Toronto, ON M4N 3M5, Canada

**Keywords:** refugee, child health, parenting, immigrant

## Abstract

Background: Families of refugee background have complex, multigenerational mental health and developmental needs that are not accounted for in current programming frameworks in Canada. Providing appropriate support services and educational resources that address the unique concerns of families of refugee background will allow for improved family cohesion and developmental outcomes for children. Parenting programs have been shown to be successful in improving parental stress, parental efficacy, and children’s mental health and well-being. This study gathers data about the experiences of caregivers of refugee background in order to develop a novel, multi-dimensional parenting program model using Community-Based Participatory Research (CBPR) principles. Methodology: This was a qualitative, CBPR study using a formative research framework. In-depth interviews (IDIs) were conducted with caregivers of refugee background and service providers that work closely with this population. Data were recorded, transcribed, and coded using deductive and inductive coding methods by two independent coders. Results: A total of 20 IDIs were conducted (7 caregivers and 13 service providers). The main topics that were identified to be incorporated into the program include: features of child development, how to address resettlement issues, child advocacy, and parenting after resettlement. Participants felt that tackling language barriers, addressing the overlapping responsibilities of caregivers attending the sessions, providing incentives, increasing awareness of the program, and using an anti-racist and anti-oppressive approach was key to the program’s success. Participants emphasized the need for trauma-informed mental health supports within the program model. Conclusion: This study describes the key considerations for a novel parenting program for families of refugee background, by engaging them as key stakeholders in the program design process. Future iterations of this project would involve a pilot and evaluation of the program.

## 1. Introduction

As of date, the world is witnessing the highest levels of human displacement. In 2022, 100 million people globally were forced to flee their homes due to conflict or persecution. Among these, nearly 27 million are refugees, 41% of whom are children under the age of 18 [1]. Currently, 60% of the world’s refugees live in urban environments [2]. Canada resettles 30,000 refugees each year [1]. Due to displacement, deprivation, economic instability, the experience of losing loved ones, suffering injuries, and witnessing violence and brutality, children of refugee background are at high developmental risk. Studies show that 10% of these children present with developmental and behavioural concerns and need support in achieving their developmental potential [3,4]. Thus, it is critical to support children of refugee background early on during their resettlement to promote positive development [5].

However, caregivers of refugee backgrounds face unique barriers in supporting their children’s development. Families potentially may encounter a range of mental health concerns associated with traumatic pre-migration events. As a result, caregivers may feel overwhelmed, disorientated, depressed, non-responsive, and have overwhelming grief that may make coping with daily tasks especially difficult [6]. Parents of refugee background may also experience diminished mental health related to post-migration stress, which may impact parenting behaviours and may inhibit the optimal development of their children [7,8].

Moreover, caregivers may face language difficulties upon resettlement. Lack of proficiency in the primary language of the country of resettlement affects all stages of healthcare access, from booking an appointment to filling a prescription, thus emphasizing the importance of supporting the language education of families of refugee background [9]. Yet, in Canada, there is a significant waitlist for refugees to attend language classes, sometimes as long as 10 months [10]. Interpretation services could be leveraged to mitigate language barriers for refugees when accessing healthcare. Literature shows that inadequate interpreter services for individuals with limited English proficiency (LEP) compromises their care, particularly for those with mental disorders [10,11]. Trained professional interpreters and/or bilingual healthcare providers positively impact the LEP patient’s quality of care, and outcomes at a feasible cost [11]. However, there is a shortage of readily available interpreters. Furthermore, language barriers exacerbate challenges to self-advocacy of refugees, difficulties in navigating their resettlement country infrastructures, and the refugee families’ economic instability due to inability to find stable employment.

The re-establishment of identity may also present a challenge to caregivers in supporting their children’s development because refugees in Canada may face many barriers in finding skilled employment, forcing many individuals to take several steps back in their careers, impacting not only their financial stability but also their self-concept and identity [6].

The unique challenge of unstable housing and frequent relocations of refugees due to national policies and regulations, have been associated with the disruption of the continuity of care for children. Finally, caregivers may have different cultural understandings of what comprises healthy child development. Since parenting practices are entrenched in cultural values, programs must include culturally sensitive adaptations in order to increase responsiveness [12].

Children with multiple and complex medical problems are especially vulnerable and can face their own unique set of challenges including transitioning back to school after a period of inconsistent education, learning a new language, coping with recent trauma, and adapting to the host community.

Parents who have generally demonstrated resilience in these emerging adversities may protect their child’s overall development [13,14]. One evidence-based practice to help mitigate these barriers are parenting programs, which have demonstrated success in improving developmental and parenting outcomes globally. This includes the Chicago Child–Parent Centers, a United States federally funded early childhood intervention program whose initial cohort of children that attended this program demonstrated positive long-term and sustained educational and social consequences [15]. Furthermore, the literature states that family-based interventions have shown to be particularly successful in refugee populations as they foster caregiver mental health while improving family functioning [16].

However, the literature surrounding supporting families of refugee background in their parenting practices as well as navigating health and education systems in their new country, remains relatively unexplored. El-Khani and colleagues reports the parenting support needs of Syrian caregivers who have resettled in shelters in Syria and Turkey [17]. Their findings cannot be extrapolated to families that resettle in Canada due to differences in language, cultural, and societal adaptations that are distinct to seeking refuge in Canada. As well, there are no existing parenting programs for families of refugee background that are designed in a participatory manner. Refugees are a diverse, heterogeneous group and variables such as their legal status, language, ethno-racial, religious, and cultural identity, all contribute to the unique challenges they may face and because of this, interventions must continuously be adapted to fit the specific needs of each group [18,19].

This study aims to identify the salient features that should be incorporated into the design of a novel parenting program that is tailored to address the unique needs of families of refugee background in the Greater Toronto Area (GTA), particularly when caring for a child with developmental or behavioural concerns.

## 2. Methods

### 2.1. Study Design

This qualitative study used a descriptive phenomenological approach. A semi-structured interview guide was developed for in-depth, individual interviews. Interviews were conducted by one of three members of the research team who have been trained in qualitative methodologies. Interviews were held at a pediatrics clinic, community centres, or over the phone. An interpreter was made available for all participants, however, only one participant requested an interpreter, and the remaining interviews were conducted in English.

### 2.2. Participants

Two groups of participants were enlisted for this research study. The first group of participants were key stakeholder groups that were identified using purposive sampling via community mapping exercises to ensure major categories of participants are represented in the study. The second group of participants represented a convenience sample from the pediatrics department of an urban academic health sciences centre and its partnering agencies and included families of refugee backgrounds and service providers that work closely with the population. Informed consent was received from all participants. Caregivers were offered grocery gift cards to honour their time in participating in their study.

### 2.3. Data Collection

Data was collected by the research team through telephone and in-person interviews between August 2019 to July 2020. The interviews lasted between 45 to 60 min and were audio-recorded and transcribed verbatim by a transcription application and then checked by members of the research team. The semi-structured interview guide was designed and utilized during the interviews to gather experiences about parenting practices from families of refugee background and key stakeholders that work closely with this population. Interviews were conducted until saturation was achieved.

### 2.4. Data Analysis

Data analysis took place alongside data collection to monitor the saturation of data. Transcriptions were hand-coded first inductively, for any patterns in participants’ experiences, and then deductively to identify categories of themes by three researchers (B.H., P.K., and R.M.). The interview transcript was imported into Dedoose for initial coding and organization of the data [20]. The transcribed texts were coded for common and contrasting themes relating to characteristics of an effective parenting intervention for families of refugee background. The research team met bi-weekly during the data analysis phase to discuss coding and analysis.

Using these themes, a description of the experiences of families of refugee background and key stakeholders who work with this population was written. The analysis focused on identifying major themes and subthemes and grounded in the interviewee’s statements. Interpretation of the data focused on both the thematic and content analysis. Thematic analysis involved an interpretation of the underlying meaning of text, and content analysis involved gathering answers to questions that were directly queried in the interviews rather than inferred in the analysis. To add to the rigour of the analysis, the authors maintained discussion notes logging the process.

### 2.5. Research Team Positionality

Due to the nature of qualitative studies, the background, experiences, assumptions, beliefs, and biases of the research team could influence the research process. To minimize this, the research team debriefed after every interview and adjusted their approach as needed. This included reframing questions or changing the location of where the interview was held to ensure inclusivity, accessibility, and comfort for participants. The interviewers all identified as people of colour who had close family members who experienced resettlement (e.g., newcomers and refugees).

### 2.6. Ethical Considerations

Research ethics approval was obtained from the Unity Health Toronto Research Ethics Board. The participants were offered full information of the study’s purpose, conduct, privacy, and confidentiality. The participants had the right to withdraw from the project at any time.

## 3. Results

### 3.1. Participants

A total of 20 individual in-depth interviewers (IDIs) were conducted. 13 of which were service providers who work closely with this population and 7 were caregivers of refugee background. 6 caregivers and 12 service providers self-identified as female (Table 1A,B). Results are also summarized in Table 2. The main themes that were found were related to program topics, barriers and facilitators and logistics.

#### 3.1.1. THEME 1: Program Topics

SUBTHEME 1: Child Development Knowledge

The caregivers expressed that they would value knowledge about the different developmental stages of children, what physical, mental, and emotional characteristics are considered normal or abnormal in each stage, how to care for a child with special needs, and what specific topics should be addressed at a given life stage.


*“I would like to understand different stages of her life, different ages.”*



*“As a parent with a child with cerebral palsy, from the beginning, we don’t know anything or we have to face everything without knowing it, but I think if we know information about my son’s case or what we have to face from the beginning, it will be more different.”*


This includes when and how to speak to their child about sexual health and reproduction.


*“My daughter asks me questions sometimes that I don’t know the answers to… or don’t know how to answer, she asked me where does a baby come from and I would like help to learn how to explain stuff that is hard for me to her.”*


Stakeholders expressed that an understanding of child development is essential, especially in the event that there is a neurodevelopmental diagnosis. Understanding typical and atypical child development and diagnoses would facilitate timely access of services.


*“I think probably for me, the biggest challenge is families not understanding the diagnosis. They are perhaps worried about being ostracized by their community...not sure how to get the services that their child needs because they really don’t understand what the disorder is.”*


SUBTHEME 2: Navigating the Healthcare System

In order to fully support the needs of their children, the caregivers believe it is necessary that they learn how to navigate the Canadian infrastructure. This includes how to secure employment, and navigating the healthcare system, especially for caregivers who need to care for their children with special needs.


*“Also, for children with? special needs. They want to know the information about how to get through the system to have their children covered.”*


SUBTHEME 3: Self-Advocacy

The caregivers expressed a deep desire to learn how to advocate for their own and their children’s rights given the inherent unfamiliarity that comes with adjusting to a foreign environment. Specifically, they would like to know what they are eligible or ineligible for, and how to obtain such services and resources.


*“There is a certain language, kind of language when you stand for your kids (...) I need to know how to get my rights and my kids rights. I don’t know how.”*


Stakeholders stated that families should be provided with the tools to advocate within schools and the health care system.


*“Learning how to advocate within the school system or within the healthcare system...as simple as learning how to go into your doctor’s appointment with your list of questions ready and feeling the confidence to get those questions asked and answered.”*


SUBTHEME 4: Caregiving Practices in the Country of Resettlement

The caregivers indicated various topics on raising a child they hope will be taught in the parenting program. The caregivers suggested that the parenting program cover how to integrate their child into a healthy lifestyle including good eating habits and engagement in physical activity.


*“They change their way of eating by style of food. Now they usually eat carbs and pizza to follow their kids, for me it’s a challenge, because it will affect their health.”*


As well, the caregivers would like to learn how to address their children when they are upset.


*“I would like to learn how I can deal with my daughter when she gets angry.”*


Finally, the caregivers would like to know how to raise good-mannered children.


*“This is the main topic for me… respectful with personality kids, it’s so important for me.”*


Stakeholders indicated that it would be important to discuss the role of child protective services because there can be misconceptions within families of refugee background as to the agencies’ role and caregivers’ rights.


*“I think when programs really explain some of the rules...what’s considered acceptable parenting and what’s not, there’s a lot of conflicts like caregivers are having with the schools and with Children’s Aid that just comes out of like cultural misunderstandings. So I think it’d be very helpful for caregivers to have this explained in their first language.”*


SUBTHEME 5: Balancing Self-Identity and Acculturation

Caregivers asserted that, while it is important for the children to integrate with the culture of their resettled environment, it is important for the children to keep their cultural roots. Thus, caregivers expressed a desire to learn how to teach their children about cultural adaptation, while also learning how to pass on their own culture and language.


*“So it’s important to learn English but it’s important to also to learn our language and our culture. So I think it’s important to the caregivers who… to know how to, to do this.”*


SUBTHEME 6: Caregivers’ Mental Health Needs

Stakeholders emphasized the need for trauma-informed mental health care because addressing these concerns will allow caregivers to engage in parenting programs.


*“If you’re a refugee, and by definition, you may have experienced some significant trauma and so working from like the mental health perspective, a lot of times when you achieve stability and safety, which is what it would be in Canada because the threat is no longer there, then you can have like a flooding of emotional experiences.”*


#### 3.1.2. THEME 2: Program Barriers and Facilitators

SUBTHEME 1: Language

Understanding that families come from various countries where the primary language is not English, caregivers indicated that language is the first barrier to address to ensure the program’s success. Specifically, the caregivers expressed the desire to learn English for benefits beyond understanding the program modules, including the ability to seek their basic needs.


*“But until they practice English, until they mastered English, the language would be a very huge barrier between them and even seek for the facilities and seek for the things they need. So the biggest challenge for me is language”*


Stakeholders echoed the concerns of language and the availability of interpretation services. This becomes difficult when trying to navigate Canadian systems.


*“Many of these families do not have English as a first language. The vast majority and depending on where you are, it can be difficult to access services particularly medical translation services, which are very different from just cultural and language translation services.”*


SUBTHEME 2: Inability to Advocate

The caregivers believe that the inability to advocate for their rights may hinder them from accessing the parenting program.


*“Here are a lot of beautiful things they [newcomers] can make use of it, but they are afraid to ask.*


Some caregivers believe that their willingness to ask for help and necessary services may be bolstered by a supportive environment fostered by stakeholders they directly engage with.


*“So also the welcoming environment for the newcomer is very important. Yeah, it will encourage him to ask about things he need to know.”*


SUBTHEME 3: Overlapping Responsibilities

The caregivers that were less likely to attend the parenting program attested to the fact that they may be occupied with time-competing responsibilities, including English classes, working, and taking care of their children.


*“You know, I would like to but right now I don’t have time because I go to school, and I have my kids.”*


Stakeholders expressed that overlapping responsibilities may deter families from attending parenting programs. They mentioned that providing childcare may encourage families to attend.


*“A lot of the families that I work with, I have dads that are working two jobs and mom has got all the kids. And so the time that might be convenient for me to run the program might not be convenient for her to attend. So ensuring there is adequate child care is important.”*


SUBTHEME 4: Awareness of Program

Many caregivers believe that a families are not aware of the various facilities that exist in Canada (e.g., YMCA, public libraries, etc.) and this pattern of unawareness may also apply to the parenting program. Therefore, caregivers can be better informed about the program through dissemination within culture-specific community offices, flyers, letters, and phone calls.


*“So most of the newcomer would even know that there is something like YMCA, like libraries, like anything. I read about it, because I know a little bit of English. And I read about it, I knew about it but for a newcomer you will not be able to discover all these facilities and all these opportunities for him and for his children.”*


The attendance to the parenting program may be further increased if when disseminating information about the program, there was an emphasis on the program’s benefits to the children.


*“To motivate people to go to the program… I think if they know for sure that they will benefit from this program and their children will benefit from this program.”*


SUBTHEME 5: Anti-Racist, Anti-Oppressive (ARAO) Frameworks

Stakeholders emphasized the need for cultural competency and anti-racist, anti-oppressive approaches within programming. Parenting frameworks in North America differ from other parts of the world and it needs to be taken into consideration.


*“There’s some cultural competence that needs to happen...what is considered culturally appropriate? We have different expectations about play, tummy time or swaddling. So I think there needs to be definitely cultural competence when thinking about parenting because we have lots of things that we do that don’t happen in other parts of the world.”*


SUBTHEME 6: Encouraging Families to Attend with Incentives

Stakeholders expressed that a facilitator to program attendance may be providing families with incentives to minimize barriers such as transportation costs and provide them with food or gift cards.


*“It’s always great to have some food and maybe offer subway or bus tokens... maybe even at first, some kind of material incentive. A gift card for their children or something else that they may need.”*


#### 3.1.3. THEME 3: Program Logistics

SUBTHEME 1: Location

In terms of where the program should be offered, the caregivers expressed a preference for a variety of locations they are already familiar with for easier access, and thus greater likelihood of attendance. This includes their local community centre, the clinic, and schools. If it’s a successful program. one participant voiced for the expansion of the program more than one location.


*“In my opinion, if it’s that good of a program, then it should be conducted everywhere.”*


SUBTHEME 2: Timing

The caregivers provided mixed responses to their preferred frequency of the program. The suggestions ranged from twice a week to once a month to ensure that the sessions are not interfering with the caregivers’ other responsibilities, and also occurring in relatively close frequency for caregivers to retain what they were taught in the previous session. In general, the sessions should be scheduled based on the group’s needs and should be a short duration program.


*“Once a week or twice a week because they will not feel like they have to change their schedule according to this one hour in a week.”*


In terms of the length of the individual sessions, most caregivers suggested one hour.


*“45 min to one hour.”*


Finally, in terms of the time of day, the caregivers preferred if the sessions occurred during the day, as their children would be in school, so their responsibilities as a parent would not compete with the attendance of the program. This timing may not be ideal for caregivers who have to work during the day.


*“For me, I think when the kids at school and it’s the perfect time for me, but for other people, maybe the timing of work or other things.”*


SUBTHEME 3: Participants

The caregivers indicated a desire for the sessions to be held in a mixed-group setting as it allows for interactions between participants, relatability and support from other caregivers on how to raise their children given their unique lived experiences as refugees, efficiency, and in general will make the program more enjoyable.


*“If you do it individually, it will take hours if you do it in a group you can do it in less time, you will be able to make people laugh and you will get answers from different caregivers from different places.”*


It will mainly be mothers that are attending the program given their husbands are the breadwinners. Ideally, the husband can attend too.


*“Yes, I would like to speak to other caregivers. My husband doesn’t have time go for work though.”*


## 4. Discussion

This study describes the key features of a parenting program tailored to the unique needs of refugee families by engaging caregivers of refugee background and services providers that work with this population as key stakeholders in the program design process.

Refugee caregivers may have difficulty supporting their children’s development due to multiple factors including experiences of distressing pre-migration events, racism and xenophobia upon arrival to their country or resettlement, language difficulties, economic instability, unstable housing and frequent relocations, lack of centralized medical and developmental services, challenges in self-advocacy, and compromised parental mental health and well-being. Having the structure, resources, and health systems to support them and their children is of critical importance as they will ultimately contribute to the national fabric via various systems (e.g., health, education, economics).

The main priorities that were identified for incorporation into the program include: features of child development, how to address resettlement issues, child advocacy, and parenting after resettlement. Participants felt that tackling language barriers, addressing the overlapping responsibilities of the caregivers attending the sessions, providing incentives, increasing awareness of the program, and using an anti-racist and anti-oppressive approach is key to the program’s success. Participants emphasized the need for trauma-informed mental health support within the program model. In order to ensure success in parenting it is essential to address the basic concerns of families. Addressing resettlement concerns such as housing, employment, legal aid, allows families to engage with programming designed to foster their child’s development. Based on the recommendations provided a program curriculum was developed (see Appendix A for sample curriculum). This program guide incorporates key priorities from stakeholders and will be revised upon further discussions and qualitative interviews. For example, since participants noted that group settings would be helpful as they’re able to interact with other caregivers, in the curriculum presented, an opportunity for caregivers to connect was scheduled and allotted. Furthermore, there is room for this program model to be adapted for each group with the use of elective, or ad hoc sessions. We recognize the heterogeneity of this population and the needs of each group of parents could vary significantly. Lastly, this model incorporates topics such as child development, mental health and well-being and addressing resettlement concerns as outlined by participants.

There is a critical gap in the literature about how to support refugee families in their parenting practices and navigation of health and education systems post-migration. Most available programming are designed with Eurocentric frameworks and perspectives around caregiving practices, which may not be relatable for many caregivers of refugee background. Participatory methodologies which include community engagement and involvement are being encouraged in health interventions and research [20]. When partnering with refugee populations, this methodology has been shown to contribute to the intervention’s success. Majority of the existing interventions and literature surrounding parenting interventions for this population were designed in a prescriptive manner and only focused on evaluation of the program. This study emphasizes the importance of collaborative approaches in program design and delivery, and it warrants the use of approaches that are trauma-informed and culturally adaptable. This study is also the first of its kind that informs programmatic and interventional design in this population in working to maximize the developmental potential of their children.

Limitations to this study include the potential challenge of generalizability to other newcomer groups. There are many classes of newcomers which include immigrants, refugee claimants, and refugees (both documented and undocumented), and their pre- and post-migration experiences may vary. Furthermore, since this study was conducted in an urban setting, the experiences of refugees in rural or other remote regions may not be consistent with the findings presented. Moreover, families were connected to an academic practice and results may differ for families that may be having challenges acquiring consistent care. Additionally, given that there were only 7 caregivers who participated, the results may not be generalizable based on these small numbers. However, these methods can be adapted and replicated with a larger sample size to further delineate the needs of this population group. Lastly, refugee groups are not homogenous. Even within similar refugee classes, their experiences may vary drastically due to different cultural, ethnic, and linguistic backgrounds and their associated socio-political contexts from their country of origin.

Future iterations of this project would involve piloting a program model based on the results of this study and undergo an iterative review process with stakeholders. Methods to leverage expertise within refugee communities will be explored. Due to the COVID-19 pandemic, virtual delivery methods of this program are being explored in order to disseminate the information in a manner that is consistent with COVID-19 guidelines around social distancing and gatherings. Once a consensus on the acceptability of the program model has been achieved, a pilot and evaluation of the program can occur.

## Figures and Tables

**Table 1 children-09-01816-t001:** (**A**). Demographic Characteristics of Caregiver Participants (n = 7). (**B**). Demographic Characteristics of Care Provider Participants (n = 13).

**(A)**		
**Characteristics of Caregiver Participants**		**N (%)**
Gender	Female	6 (85.71%)
	Male	1 (14.28%)
Age of Caregivers (Years) *	-	41.28 (4.46) *
Education	Secondary School	3 (42.85%)
	College/University	1 (14.28%)
	Post-Graduate	3 (42.85%)
Home	Home/Condo (Rented)	2 (28.57%)
	Apartment	4 (57.14%)
	Shelter	1 (14.28%)
Number of Children	1	1 (14.28%)
	2	3 (42.85%)
	3	2 (28.57%)
	4+	1 (14.28%)
Individuals Providing Care	1	2 (28.57%)
to Child at Home	2	5 (71.42%)
Relationship of Individuals	Mother & Father	4 (57.14%)
Providing Care to Child at	Mother & Grandmother	1 (14.28%)
Home	Mother	2 (28.57%)
Current Occupation	Employed	1 (14.28%)
	Unemployed	6 (85.71%)
Source of Income	Full Time Work (Spouse)	3 (42.85%)
	Ontario Works	4 (57.14%)
Country of Origin	Syria	1 (14.28%)
	Sudan	1 (14.28%)
	India	1 (14.28%)
	Egypt	1 (14.28%)
	Iraq	1 (14.28%)
	Uganda	1 (14.28%)
	Ethiopia	1 (14.28%)
**(B)**		
**Characteristics of Care Provider Participants**		**N (%)**
Gender	Female	12 (92.3%)
	Male	1 (7.69%)
Number of Years Working	0–5	4 (30.76%)
With Families of Refugee	6–10	2 (15.38%)
Background	11–15	3 (23.08%)
	16–20	2 (15.38%)
	20+	2 (15.38%)
Professional Designation	Early Childhood Educator	1 (7.69%)
	Nurse	1 (7.69%)
	Child & Youth Worker	1 (7.69%)
	Settlement Counsellor	2 (15.38%)
	Social Worker	3 (23.08%)
	Developmental & Behavioural Counsellor	1 (7.69%)
	Physician	1 (7.69%)
	Program Lead	1 (7.69%)
	Patient Navigator	1 (7.69%)
Type of Employer	Community Health Centre	3 (23.08%)
	Hospital	5 (38.46%)
	Community Agency	4 (30.76%)
	Not-for-Profit Organization (NPO)	1 (7.69%)

* Presented as Mean (SD).

**Table 2 children-09-01816-t002:** Summary of Themes.

Themes	Subthemes	Quotes
1. Program Topics	1. Child Development Knowledge	*“I would like to understand different stages of her life, different ages.”*
	2. Navigating the Healthcare System	*“Also, for children with special needs. They want to know the information about how to get through the system to have their children covered.”*
	3. Self-Advocacy	*“There is a certain language, kind of language when you stand for your kids (...) I need to know how to get my rights and my kids rights. I don’t know how.”*
	4. Caregiving Practices in the Country of Resettlement	*“I think when programs really explain some of the rules...what’s considered acceptable parenting and what’s not, there’s a lot of conflicts like parents are having with the schools and with Children’s Aid that just comes out of like cultural misunderstandings. So, I think it’d be very helpful for parents to have this explained in their first language.”*
	5. Balancing Self-Identity and Acculturation	*“It’s important to learn English but it’s important to also to learn our language and our culture. So, I think it’s important to the parents to know how to do this.”*
	6. Caregivers’ Mental Health Needs	*“If you’re a refugees, and by definition, you may have experienced some significant trauma and so working from like the mental health perspective, a lot of times when you achieve stability and safety, which is what it would be in Canada because the threat is no longer there, then you can have like a flooding of emotional experiences.”*
2. Program Barriers and Facilitators	1. Language	*“But until they practice English, until they mastered English, the language would be a very huge barrier between them and even seek for the facilities and seek for the things they need. So the biggest challenge for me is language”*
	2. Inability to Advocate	*“Here are a lot of beautiful things they [newcomers] can make use of it, but they are afraid to ask.*
	3. Overlapping Responsibilities	*“A lot of the families that I work with, I have dads that are working two jobs and mom has got all the kids. And so, the time that might be convenient for me to run the program might not be convenient for her to attend. So, ensuring there is adequate childcare is important.”*
	4. Awareness of Program	*“To motivate people to go to the program… I think if they know for sure that they will benefit from this program and their children will benefit from this program.”*
	5. Anti-Racist, Anti-Oppressive (ARAO) Framework	*“There’s some cultural competence that needs to happen...what is considered culturally appropriate? We have different expectations about play, tummy time or swaddling. So, I think there needs to be cultural competence when thinking about parenting because we have lots of things that we do that don’t happen in other parts of the world.”*
	6. Encouraging Families to Attend with Incentives	*“It’s always great to have some food and maybe offer subway or bus tokens... maybe even at first, some kind of material incentive. A gift card for their children or something else that they may need.”*
3. Logistics	1. Location	*“In my opinion, if it’s that good of a program, then it should be conducted everywhere.”*
	2. Timing	*“Once a week or twice a week because they will not feel like they have to change their schedule according to this one hour in a week.”*
	3. Participants	*“If you do it individually, it will take hours if you do it in a group you can do it in less time, you will be able to make people laugh and you will be get answers from different parents from different places.”*

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
