# Peer review of "For the Parent, by the Parent: Creating a Program to Empower Parents of Refugee Background in Canada Using Novel Participatory Approaches"

_children, 2022, doi:10.3390/children9121816_

Round 1

Reviewer 1 Report

This is an interesting and well-written paper. However it has a very scant review of existing literature in this area and only cites one paper on parenting practices of refugees sheltering in Turkey and Syria. In addition, most of the cited references seem to be Canadian. As far as I am aware other refugee resettlement contexts (e.g. Europe, USA, Australia and NZ) would have studies exploring this territory against which the findings of this study could be compared and contrasted. This would potentially strengthen the Discussion which is currently not adding much to what I perceive is an existing base of evidence. The data is presented logically but there is little insight to be drawn from the paper overall due to limitations in the Introduction and Discussion.

Reviewer 2 Report

An interesting study but I have a number of concerns with it that include the methodology, analysis, and presentation of the findings and conclusions.

You start by noting that the world as a whole is experiencing its highest "human displacment" and make reference to data that is seriously outdated. While the point that you make is correct it is based on data that is very outdated, 2018. The latest figures from UNHCR indicates that are now over 100 million people who are forcibly displaced. The Ukraine War is the principal reason why the figures are so high, but, it is also due to the number of armed conflicts that are taking place in the world today, more than any time since WW II. Further, in 2018 perhaps half of those who were forcibly displaced were children but now the figure stands at 41 percent. You should be using the latest statistics available for the number of the world's refugees and others who are forcibly displaced. 

Turning to methodology, it is evident that you are quite aware of the limitations of your research study based on only seven interviews with caregivers and 13 service providers. Twenty indepth interviews with a mixed group of respondents. I find this not only unrepresentative but hardly adequate for a number of reasons. First, Toronto is undoubtedly the principal gateway for the reception of those seeking asylum in Canada. The country as a whole receives about 25,000 or more asylum seekers on average a year and many of these end up being settled in Toronto. I am sure these figures are available. And to have only interviewed seven refugee caregivers/people seems woefully inadequate, even if they were indepth interviews that lasted about one hour. Likewise, for the service providers. Even out of the NGO community of service providers for refugees in the City of Toronto this would be highly unrepresentative of the numbers of service providers who work with refugees. The results coming from such a small sampling of service providers would be extremely limited at the very best.

Moreover, it is worth noting that the number of undocumented people who are in Canada creates a further methodological concern. Many of these are likely forced migrants and because Toronto and other cities across Canada are now "sanctuary" cities are likely receiving social services but do not enjoy official immigration status in Canada. This adds a further complicating factor that is unacknowledge in your study.

Combining the results both the refugee caregivers and the service providers seems problematic in a number of respects. Clearly, they are different both functionally and also organizationally, the first as family members, and the second as employees working within a social service agency. The experiences of the former could clash with the later. For instance, I am a service provider but have to deal with a difficult client. Or I am a caregiver who feels that my service provider is ignoring me and not providing me with all the services I need. 

With respect to the analysis of the presentation of the findings, I found the presentation of the various themes and subthemes and quotations to be rather artificial to some degree. There is an explanation of how you identified the themes and various subthemes but my impression is that the quotations presented are merely consistent with these but do not present any indication of how prevalent they are or were among the 20 persons interviewed. There is no sense of how many caregivers or service providers said what when they were asked a particular question in their interviews. How did the two differ, if at all, in their responses to the questions in your interview instruments?

Further, in this regard, there was no overriding sense of a clear pattern in the way you presented the interview data. Yes, thematic areas and subthemes within areas were identified and presented with sample quotations. But what do you make of this? What are the empirical findings that emerge from these interviews. I suspect that they are very different for the refugee caregivers and the service providers.

While intuitively I accept all of the arguments made regarding the difficulties refugees face in terms of their resettlement in a new country where the language, culture, and environment is very different from their previous life experience, not to mention the trauma they may have suffered in their countries of origin or en route to their destination country. In this regard, I noted with interest that the seven people interviewed all came from either a Middle Eastern, African, and one Asian country. All these countries have experienced serious armed conflict situations that are either ongoing or existed in the past. What was interesting is you selected service providers who came from a family refugee background. But there is no indication what this was or whether they may have come from these same seven source countries.

Finally, I found the conclusions quite general and outlined the limitations of the study and avenues for future research. But I did not find any definitive statements of any findings from the interviews that could be used for designing a program for addressing the immediate or long-term developmental needs of refugee families and their children. There was reference with respect to possible take up and the time of day when it should be offerred, etc., but what would be the essentials for such a program? The conclusions were rather quiet in this regard. 

I found errors in the text at lines immediately above 165 and at 385.

Round 2

Reviewer 1 Report

I wonder if the literature review could include some additional insights and if that would assist in developing a more insightful conclusion on what's different in what you found in this study and what's similar to other studies. In my professsional experience I have come across research that has looked at parenting interventions (e.g., Renzaho and Vignevic, 2011) and I wonder/query whether your search of the literature has been strong enough in order to compare and contrast your findings. 

Author Response

I wonder if the literature review could include some additional insights and if that would assist in developing a more insightful conclusion on what's different in what you found in this study and what's similar to other studies. In my professsional experience I have come across research that has looked at parenting interventions (e.g., Renzaho and Vignevic, 2011) and I wonder/query whether your search of the literature has been strong enough in order to compare and contrast your findings. 

Thank you for your comment. The discussion has been edited to include the fact that other programs focused on evaluation, rather than design and delivery. This inclusivity piece is what we emphasized to be different from other studies. 

Reviewer 2 Report

I appreciate the changes made to the submission but I am afraid that I have to take issue with a number of points in your response and in your current text.

-- A qualitative study based on seven refugee parents with children out of a sizeable potential population of thousands of refugees who resettle in Canada each year is suggestive at best and not at all generalizable. This needs to be emphasized. I understand that this is being used for designing a pilot for a larger study but the use of only seven interviews with refugee parents is quite limiting in itself.

-- The same applies with those who provide services to refugees and their families when you have only interviewed 13 people out of a total population in Toronto alone that is in the hundreds, if not thousands.

-- Apologies for any misunderstandings on my part, but to avoid this with other readers I might suggest that you consider rewording line 127 so that it is clearer.

-- The supplementary table is helpful but it requires a title and perhaps a brief summary.

This may only be me, but I would be cautious in the use of the term "newcomer" and mixing immigrants with refugees who have been forcibly displaced or forced to migrate. Immigrants, more often than not, are voluntary migrants who, yes, are generally seeking a better life, new opportunities, the opportunity to study abroad, or simply new adventures. Refugees are fleeing persecution, the severest violations of their most fundamental human rights - life, liberty, and security. Your study has been limited to interviewing refugees with children and those who provide services to refugee families. I would avoid mixing the two migrants: immigrants and refugees.

With respect to the different types of refugees who resettle in Canada, as you are no doubt aware there are those who arrive in Canada and claim refugee status and those who are resettled to Canada from various countries after being determined to be refugees by UNHCR. Refugees who are resettled to Canada can be government sponsored, privately sponsored, or blended visa-officer referred (BVOR). You do mention that there is a variation in the refugees in your submission, but it is unclear whether you intend to distinguish between these various types of refugees. One might expect that there could be a variation between those who arrive here and claim asylum versus those refugees who are resettled from abroad. The major difference is, of course, those who arrive here on their own and claim asylum must go through the refugee hearing process to determine if they are indeed refugees. Whereas those who are resettled from abroad have already been determined to be refugees before their arrival. Whether there is a substantial difference between the various types of refugees has been a subject of much research, discussion and debate in the literature, especially, between government and privately sponsored refugees. 

At line 418, the word "be" is missing before "generalizable," as in " ... may not be generalizable based on ..." Proofread the submission with care.

Author Response

I appreciate the changes made to the submission but I am afraid that I have to take issue with a number of points in your response and in your current text.

-- A qualitative study based on seven refugee parents with children out of a sizeable potential population of thousands of refugees who resettle in Canada each year is suggestive at best and not at all generalizable. This needs to be emphasized. I understand that this is being used for designing a pilot for a larger study but the use of only seven interviews with refugee parents is quite limiting in itself.

-- The same applies with those who provide services to refugees and their families when you have only interviewed 13 people out of a total population in Toronto alone that is in the hundreds, if not thousands.

Thank you for your comment. In lines 431 and 432 we have mentioned that the small numbers are a limitation in this study.

-- Apologies for any misunderstandings on my part, but to avoid this with other readers I might suggest that you consider rewording line 127 so that it is clearer.

Thank you for your comment, 127 has been reworded.

-- The supplementary table is helpful but it requires a title and perhaps a brief summary.

A title has been added, the manuscript includes a description of the table contents.

This may only be me, but I would be cautious in the use of the term "newcomer" and mixing immigrants with refugees who have been forcibly displaced or forced to migrate. Immigrants, more often than not, are voluntary migrants who, yes, are generally seeking a better life, new opportunities, the opportunity to study abroad, or simply new adventures. Refugees are fleeing persecution, the severest violations of their most fundamental human rights - life, liberty, and security. Your study has been limited to interviewing refugees with children and those who provide services to refugee families. I would avoid mixing the two migrants: immigrants and refugees.

This has been clarified within the manuscript by ensuring that we minimized the use of the word newcomer. The word newcomer now only will appear in the interview quotes as the participants described their experiences.

With respect to the different types of refugees who resettle in Canada, as you are no doubt aware there are those who arrive in Canada and claim refugee status and those who are resettled to Canada from various countries after being determined to be refugees by UNHCR. Refugees who are resettled to Canada can be government sponsored, privately sponsored, or blended visa-officer referred (BVOR). You do mention that there is a variation in the refugees in your submission, but it is unclear whether you intend to distinguish between these various types of refugees. One might expect that there could be a variation between those who arrive here and claim asylum versus those refugees who are resettled from abroad. The major difference is, of course, those who arrive here on their own and claim asylum must go through the refugee hearing process to determine if they are indeed refugees. Whereas those who are resettled from abroad have already been determined to be refugees before their arrival. Whether there is a substantial difference between the various types of refugees has been a subject of much research, discussion and debate in the literature, especially, between government and privately sponsored refugees. 

Thank you for your comment, due to the small sample size we will not be able to delineate based on subclasses of refugees. This is something we hope to do in future studies examining the program model.

At line 418, the word "be" is missing before "generalizable," as in " ... may not be generalizable based on ..." Proofread the submission with care.

Thank you for the correction.